# Position: Agentic Systems Should be General

**Elron Bandel** [1]   **Asaf Yehudai** [1]   **Alexandre Lacoste** [2]   **Avijit Ghosh** [3]   **Graham Neubig** [4]   **Margaret Mitchell** [3]
**Michal Shmueli-Scheuer** [1]   **Leshem Choshen** [1 5]

## Abstract

We call for the development of agentic systems that thrive in new environments. Agentic systems, comprising foundation models, tools, and an execution strategy, have demonstrated strong capabilities, yet their development is often constrained by narrow benchmarks and their operation is siloed to limited environments.

This paper advocates for developing general, adaptive agents that excel across diverse environments, from terminals and web interfaces to biological and embodied settings. We examine current limitations, explain the potential of increased generality, and identify immediate development priorities. Finally, we argue that protocols and evaluation must prioritize adaptiveness to foster a shared ecosystem for general-purpose agentic systems.

## 1. Intro

In recent years, we saw a rise in the generality of AI solutions, LLMs moved from domain-specific models to general-purpose ones (Raffel et al., 2020), and broadened to incorporate reasoning capabilities (Guo et al., 2025). A significant step in this direction is the transition to agentic systems that tackle complex, long-horizon tasks that require repeated interactions with the external environment. Yet, most of these agents are developed, improved, and evaluated in domain-specific settings (see §2).

We propose that agent developers should aim for generality. Where generality refers to minimizing human effort in new settings. Thus, we see domain-specific agents as an intermediate stage in the development of agents and envision that every next generation of agents should and will become more general and unified. We term these agents

general-purpose agents, possessing the ability to perform different types of agentic tasks across unfamiliar and varied environments.

Achieving such generality requires a collective effort to build a general-purpose agentic ecosystem. This necessitates benchmarks that reward, rather than penalize, generality, and protocols that provide agents with essential environmental context.

Generalization is a goal more than a predefined set of requirements. As such, an agentic system that merely fulfills two tasks in the health domain is more general than if it only performs one. Thus, we do not expect to reach Agentic General Interaction.

Still, we aim to highlight the directions in which agentic systems are commencing to generalize today and might master in the foreseeable future. Among those are performing different tasks with new tools, across domains and environments, using different modalities, embodiments, and complex data geometries. In all of those, we should strive to reduce human intervention per instance.

General agents represent a promising frontier because they consolidate fragmented research efforts, ensuring that breakthroughs benefit the entire field rather than being lost in isolated sub-communities or redundant studies. By supporting diverse signals, these agents develop more robust, adaptable solutions compared to narrow systems. Furthermore, the versatility of a general-purpose architecture allows for significant impact across vastly different use cases, while simultaneously reducing the cost and complexity of implementing the technology for new, specific applications.

> **The Core Position**
>
> When developing agentic systems, the community should favor **general-purpose designs** that adapt across environments over narrow, benchmark-specific solutions. This includes building the **protocols, evaluation frameworks, and development practices** needed to support adaptable agents that minimize per-deployment human effort.

[1]IBM Research  [2]ServiceNow Research  [3]Hugging Face  [4]Carnegie Mellon University  [5]MIT. Correspondence to: Elron Bandel <Elron.Bandel@ibm.com>, Asaf Yehudai <Asaf.Yehudai@ibm.com>, Leshem Choshen <leshem.choshen@mail.huji.ac.il>.

*Proceedings of the 43$^{rd}$ International Conference on Machine Learning*, Seoul, South Korea. PMLR 306, 2026. Copyright 2026 by the author(s).

## 2. What are Agentic Systems

Agentic systems are pipelines that sustain interaction with an environment to perform a task, integrating components such as foundation models, tools, and human intervention through various activation patterns (e.g., Yao et al., 2022). We now define key terms related to agentic systems that recur throughout the paper.

### 2.1. Definitions

**Agentic System** or **Agent** is a system that performs actions and observes their consequences in pursuit of a task. In our context, agentic systems are multi-component frameworks, specifically ones including a foundation model. We define those components next.

**Environment** is the external world with which an agentic system interacts: it provides observations to the agent and is modified by the agent's actions, typically via tools.

**Agentic Pattern** is the core algorithm of an agentic system orchestrating its components, such as foundation models, memory mechanisms, context compression, and tool execution. Given initial input, the agentic pattern governs how these components interact and react to one another (e.g., Yao et al., 2022; Wang et al., 2024).

**Foundation Model** is a learned model capable of performing diverse tasks in a certain domain. In contrast, a **tool** is any code that is available to the agent to manipulate information or interact with the environment or the system's state (Schick et al., 2023). While a foundation model can be seen as a tool, it often plays a unique role in the agentic-pattern's orchestration and decision-making, and we hence separate it for ease of discussion.

In terms of generality, we see the foundation model and tools as separate from the agentic pattern, but some patterns do currently rely on specific models. It remains to be seen whether these will ultimately be separate from the system, inseparable, or a hybrid, where some tools and foundation models are provided to the agent in new settings, while others ship as part of the general solution.

**Protocol** defines the standard for exchanging dynamic information to and from the agentic system. This includes environmental state, task specifications, and external integration. For example, the Model Context Protocol (MCP) (Anthropic, 2024) standardizes how agents access data sources, tools, and workflows to retrieve information and execute tasks (MCP Project, 2025).

## 3. What are General-purpose Agents

We define a more general agentic system as one that can **successfully act across a broader range of scenarios**. This

adaptability may stem from the agent's ability to recognize novel situations through interaction or from explicit communication of environmental features.

An equivalent perspective sees **generality as the reduction of manual effort**. The more general a system is, the less effort is invested in every new situation. Less effort to develop a new agent or configure an existing one to work with new tools and environments.

Notably, generality is not a binary property but exists as a spectrum: agents differ in the breadth of environments and novel settings they can handle.

An accompanying goal to generality is always **performance**. The most general agentic system is one that does nothing in every situation. This extreme example serves to remind us that when we describe the research field, we call for more general systems and ecosystems to be developed, but among them, those systems should become more competent in the general settings they put forth.

**General Performance** highlights the gap between theoretical possibility and practical execution. While an agent equipped with minimal general-purpose tools or in open environments may be capable in principle of performing any task, analogous to Turing-Completeness, in practice, such capability may be inefficient or unreliable. An agent could, for instance, construct its own tools given access to code execution or a general computing interface, yet the resulting performance may be prohibitively costly. Thus, for example, a "generalist agent for the web" (Deng et al., 2023) can still be more general by interacting with other environments such as terminals. Overall, we discuss practical generality rather than theoretical possibility.

## 4. The Current Spectrum of Generality

Agentic research has rapidly expanded, producing systems that vary widely in their scope, assumptions, and claims of generality. We identify that contemporary agent logic is often tied to specific environments, leaving clear avenues for future research (Yehudai et al., 2025). Similarly, agentic benchmarks (e.g., Barres et al., 2025; Trivedi et al., 2024; Chae et al., 2024; Yang et al., 2025) don't measure general abilities, which we discuss in §8.

In this section, we first outline broad categories of agent generality, then examine where benchmark-conditioned agents that are fitted for specific environments or domains fall short, and finally highlight environment-agnostic designs as a more robust path toward general agents.

### 4.1. Levels of Agent Generality

A variety of work focuses on building agentic systems for a specific task or domain, either to showcase a general method

or because the task itself is of intrinsic value. Examples include agents specialized for web tasks (Bohra et al., 2025), medical reasoning (Zhou et al., 2023), or legal workflows (Tang et al., 2023b). Such systems do not aim for broad generality, instead prioritizing performance and robustness within a fixed domain.

In contrast, research on subcomponents focuses on versatile capabilities, including unseen tool use (Tang et al., 2023a), performing multiple tasks within a single domain (Wu et al., 2024), and agentic behaviors such as memory, self-improvement, and reasoning (c.f., Luo et al., 2025). These methods rely primarily on a language backbone and are largely domain-agnostic, making them applicable across a wide range of agentic systems.

Finally, some foundation models (famously, GATO; Reed et al., 2022) were among the first to claim broad generality. Models such as GATO can be viewed as a single-component agentic system in their own right, operating across many environments, though adapting to new ones often requires additional training. While foundation models may continue to grow more general, our focus here is on multi-component agentic systems, which remain central to contemporary agent research.

Importantly, the generality of a model or pattern does not automatically ensure generality at the system level. We now examine how this gap manifests in practice.

### 4.2. Benchmark-Conditioned General Agents

Supporting multiple environments is a key step towards generality. However, common approaches tend to encode environment-specific assumptions directly into the agent logic, limiting their applicability to unseen environments.

**HAL Generalist Agent.** The HAL Generalist Agent (Kapoor et al., 2025b) exemplifies this pattern. While HAL supports multiple environments, it behaves differently depending on the benchmark used to measure its capabilities. For example, the implementation contains more than 20 instances of conditional logic of the form `if kwargs['benchmark_name'] == ...`, which are used to encode assumptions about dataset structure, available tools, and task-relevant information that are specific to individual benchmarks. For example:

```
elif kwargs['benchmark_name'] == 'scienceagentbench':
    DATA_INFO_PROMPT = """You can access the dataset at
        '{dataset_path}'. Here is the directory
        structure of the dataset:
'''{dataset_folder_tree}'''
Here are some helpful previews for the dataset file(s):
{dataset_preview}"""
```

**CUGA: Configurable Generalist Agent.** CUGA (Marreed et al., 2025) similarly supports multiple benchmarks

through explicit configuration. While it is not tied to a single task or domain, it requires manual, benchmark-specific setup to function correctly. Notably, CUGA includes dedicated infrastructure for AppWorld, such as the `appworld_auth_manager.py` module, which automatically logs into applications on behalf of the agent, and the system prompt instructs it to assume an authenticated status. Consequently, while HAL and CUGA are configurable, they remain constrained in novel domains.

### 4.3. Environment-Agnostic Agent Design

In contrast to benchmark-conditioned systems, some agents adopt an environment-agnostic design that avoids embedding environment-specific assumptions into their pattern. `mini-swe-agent` (Yang et al., 2024), though not claiming generality, epitomizes many key properties of generality.

The core loop of `mini-swe-agent` is environment-agnostic and interacts exclusively through a generic command-execution interface (`env.execute`), driven by model-produced `bash` code blocks. The agent does not branch on benchmark identifiers or include environment-specific handlers. Instead, it follows a minimal and reusable protocol: (i) render task and context templates, (ii) query the model, (iii) parse a single CLI action, (iv) execute it, and (v) feed the observation back to the model, with well-defined error recovery for format errors, timeouts, and step or cost limits.

This design yields a clear extension mechanism: supporting a new benchmark requires only implementing an `Environment` backend exposing `execute()` and the necessary template variables, without modifying the agent's control flow. As long as a target environment provides a CLI, or can be wrapped as one, `mini-swe-agent` can, in principle, operate across unseen environments without prior benchmark knowledge baked into its core logic.

## 5. Sparks of General Agents

If environment-agnostic design truly captures agent generality, it should lower engineering complexity and reduce operational cost while maintaining competitive performance. We examine empirical cases where general agents have already shown promise. Though currently rare and hence anecdotal, these examples highlight potential advantages of generality.

### Case 1: Scientific Agents

ASTA Bench (Bragg et al., 2025), an effort towards benchmarking deep scientific research agents, provides a clear test. As shown in Table 1, the specialized ASTA-v0 system scores 55% at $3.40 per task and contains subsystems exceeding 13,000 lines of code (LOC).Yet the second-best

system is a 300-line ReAct general agent scoring 44% at just $0.31. On the literature-understanding subtask, ReAct scores 53%; although still below the Asta agent (62%), outperforming both the specialized ASTA Paper Finder (21%) and OpenAI Deep Research (19%).

### Case 2: SWE Agents

In SWE-Bench (Yang et al., 2025), the specialized SWE-Agent scores 67%, but the tiny, domain-agnostic Mini SWE-Agent scores 65% while being 30 times smaller and about 7 times cheaper per run (Table 2). Across both scientific and SWE settings, small general agents of a few hundred lines consistently achieve 70% to 95% of the performance of systems that are thousands of lines.

## 6. Supporting Viewpoints

Progress in Machine Learning and computing has repeatedly (though not monotonically) evolved toward higher levels of abstraction. Heuristics gave way to algorithms; algorithms to learning systems; and per-task models to foundation models trained on broad data distributions. Each transition reduced *coupling between problems and solutions*, shifting the primary bottleneck from per-task manual engineering toward preparation through data, compute, and scalable optimization. Agentic systems extend this trajectory by abstracting over interaction itself: rather than mapping fixed inputs to outputs, agents define strategies, selecting actions that shape future observations across environments and tools.

Based on this historic trend, and the specifics of agentic systems, we discuss the potential benefits of developing general agentic systems in this section and alternative viewpoints in §7.

**Better Performance**  A consistent lesson from this history is that studying general mechanisms creates stronger tools. Both in computer vision (Krizhevsky et al., 2012) and natural language processing (Devlin et al., 2019), hand-engineered pipelines were displaced by neural networks trained on large datasets, which work better, require less human effort, and are less brittle. As articulated by the "Bitter Lesson," approaches that rely on scalable computation, data and learning tend to eventually outperform those that require manual effort (Sutton, 2019).

**Robust**  Occam's Razor is often taken out of its original context and applied to machine learning, where a repeated finding in theory and practice is that the solution with the fewest assumptions (and sometimes degrees of freedom) works best. Such solutions generalize better, are more robust, and less prone to overfitting (Shalev-Shwartz & Ben-David, 2014; Valiant, 1984; Gokhale et al., 2022). Specifically, unlike domain agents that can be over-specialized

and tailored to a specific task or domain, general agents work with different environments, forcing them to generalize. They receive diverse signals from a wide range of environments, requiring them to be robust.

**Evaluation**  As agents become capable across more domains, they can be evaluated on a wider and more diverse set of tasks, as some suggest (Bandel et al., 2026; Lacoste et al., 2026) . Although broad evaluations may not fit any single practitioner, they are more likely to cover a task's requirements and test relevant capabilities. In contrast, evaluations tied to specific agent settings are less transferable, since practical use cases vary substantially.

**Centralizing Efforts**  Generality is as beneficial for the field as it is for the quality of the designed systems. Focusing on general-purpose systems unifies an otherwise fragmented research landscape. By moving away from isolated use cases, we eliminate the "redundancy trap" where researchers independently rediscover the same fundamental principles, as has been reported for engineering (Gamma, 1995), Machine Learning (Sculley et al., 2015), and science in general (Kuhn & Hacking, 1970). Furthermore, a generalist framework cuts the communication overhead required to share insights across subfields. This alignment of communication, goals, and solutions will likely show its gains in speed and progress.

**Reducing Entry Barriers**  Generality can support a broader community of users. Under our definition, generality emphasizes minimizing human effort when adapting to new scenarios, rather than requiring a single system to work in all cases. Thus, the expertise required for tweaking is also expected to be reduced. This trait was often acclaimed as a main factor for the speed of innovation in the Machine Learning field and a reason to call for open solutions (Sonnenburg et al., 2007; Hosanagar & Saxena, 2017; Birhane et al., 2022; Manchanda et al., 2024) and we hope that with lowering other barriers to entry like compute, it will lead to it here as well.

**Better starting point for Specialized Agents**  By our definition, general agentic systems are capable of performing a wide range of tasks with minimal human guidance, thus providing a strong starting point for specialization. With general reasoning, adaptation and cross-domain needs handled, developers can focus on high-value, domain-specific challenges, accelerating innovation. This pattern mirrors the historical trajectory of pretrained models, which, once trained on broad corpora, could be efficiently fine-tuned for specialized NLP tasks (or create a dedicated agentic pipeline), dramatically reducing development time. Consequently, investing in general agentic systems is likely to yield superior starting points for specialized agents, both in

| Agent | LLMs used | ASTA score | Cost per task | LOC |
|-------|-----------|------------|---------------|-----|
| ASTA-v0 | Claude 4 Sonnet, Gemini 2.5 Flash, O3, GPT 4.1, GPT-4o | 53% | $3.40 | $> 13,768$ |
| ReAct | GPT-5 | 44% | $0.31 | 358 |

*Table 1.* In scientific tasks, the generalist agent ReAct approaches ASTA-v0's performance with far less cost and complexity

| Agent | LLM | SWE-Bench score | Cost per task | LOC |
|-------|-----|-----------------|---------------|-----|
| SWE-Agent | Claude 4 Sonnet | 67% | $\sim$$2.50 | 4,161 |
| Mini SWE-Agent | Claude 4 Sonnet | 65% | $0.37 | 131 |

*Table 2.* In software engineering tasks, a minimal general agent nearly matches a specialized agent while being far smaller and cheaper.

efficiency and performance.

**Interoperability through decoupling (Narrow Waist)**
At the system level, history suggests that generality improves robustness by reducing brittleness at interfaces. As systems scale, failures increasingly arise at the boundaries between components rather than within individual modules. The Internet's success illustrates this principle: a narrow, stable waist in the protocol stack enabled innovation above and below it while preserving interoperability (Clark, 1988; Cerf & Kahn, 1974). Agentic systems are similarly composite, integrating models, tools, environments, memory, and control logic. General agentic abstractions, such as an environmental information protocol, can serve as a "narrow waist" decoupling the developments of tools, models and agentic patterns.

**Widespread Gains** With uses in more settings, more research, and more users, agentic systems are likely to be deployed more. When those solutions also share many of the same components, the result is clear. New improvements impact a wider audience.

**Accountability through visibility** So far, we discussed the potential of agentic systems for increased use. We now turn to their potential in mitigating risks. Accountability through visibility is the benefit that emerges when general-purpose agents are widely used and openly scrutinized. Broad exposure invites continuous examination, criticism, and defense, making failures harder to hide and accelerating understanding of safety and moral risks. This shared scrutiny distributes responsibility across the community, strengthening both accountability and ethical oversight over time. Some also claimed that the robustness that comes from general testing would induce safety (Jin & Lee, 2025).

**Governance** Standard centralized bottlenecks in systems allows for simple intervention. Thus, for example, to avoid certain agentic patterns, there is a clear point of intervention in the general agent, the communicating layers, standards,

code or developments may enforce not leaning towards such unwanted behaviors. Such centralization points also aid regulatory bodies to audit on the whole ecosystem at once (Dafoe, 2018).

Taken together, these precedents motivate prioritizing general agentic systems over narrowly engineered alternatives. Agents operate in regimes characterized by high task diversity, long horizons, evolving environments, and limited opportunities for manual tuning—conditions under which general abstractions have repeatedly proven effective. While generality does not guarantee optimal performance on every task, the history of abstraction in computing suggests it is a favorable strategy for building agentic systems and infrastructure intended to operate broadly, adapt continuously, and minimize ongoing human intervention.

## 7. Alternative Views

We provide multiple adjacent viewpoints that our claim raises as a way to better understand the nuances of what we call for.

**Agentic systems should be specialized.** The most direct alternative view to the current work is that agentic systems should *not* be general; they should remain constrained to specific tasks and environments. This has a few potential benefits:

- *Control.* Using explicitly defined tasks and domains of operation enables clarity on what specifically an agent should and should not do, and provides insight into potential beneficial and problematic outcomes of agent operation. As such, developers can adopt a high degree of precision in limiting harmful agent functionality, while enabling task-specific supportive actions. This can help to appropriately address multiple foreseeable events stemming from an agent's execution, including helping to prevent safety and security issues (He et al., 2025; Sheggam & Zhang, 2024). In this paper's position, many of the benefits of control still remain with general agents: One can limit foreseeable problematic

agent operations, for example, by constraining its environment and tools. However, we acknowledge that it is harder to control the effects of the agentic pattern, and correspondingly, would have less insight into how the agent might interact with the world. We argue that the benefits of generality are worth this cost.

- *Predictability.* Related to the above, specialized agents can be more predictable. In this view, generality vs. predictability might be seen as two views of the same potential: Just as unforeseen opportunities and assistance from AI agents are enabled by generality, so too are unforeseen harms. This is akin to a "glass-half-empty" vs. "glass half full" argument.

- *Efficiency.* When the domain of operation for an agent is explicitly defined, it can be highly optimized, cutting unnecessary abilities, data consumption, and compute, making the extra human effort worthwhile. We argue that this efficiency will be available to those with resources, but it would raise the barrier of entry for the rest (§6).

- *Evaluation rigor.* The more specialized the system, the clearer it is what to evaluate, tying agents to their use. We argue that narrow specifications are easier to game and overfit, which are likely to lead to approaches that only work well in the setting of a specific target benchmark.

**General ability might enable increased autonomy, which can bring unforeseen risks.** While the autonomy of AI agents entails decreased need for human *development* per task, there is a worry that this will also mean less control or manual intervention points during agent *operation*. We argue that generality is a distinct property from autonomy, and the two concepts should not be conflated: Concern about autonomy does not entail diminishing generality. Such a position would need to be rigorously examined in its own right. We refer interested readers to Feng et al. (2025); Mitchell et al. (2025); Kasirzadeh & Gabriel (2025) for details on the interplay of autonomy and safety.

**Specialized agents are good.** This is a statement we hold alongside our stated position. The two positions are not mutually exclusive; solving common problems is likely to create the best building blocks for building specialized agents where those are worth the extra effort, and might solve out-of-the-box other cases as well.

**The foundation models should be general, not the agent.** This view holds that the models underlying the agents are still getting better, and incorporate more abilities, perhaps reaching the generality criteria themselves. We do not focus here on whether more effort should be invested in agents or models. Instead, we argue that, as agents are already being studied, it is useful to focus increased effort on general ones.

**General agentic systems can be harder to debug, or prone to compounding errors, compared to narrowly engineered solutions.** Indeed, general mechanisms often underperform specialized ones on isolated benchmarks with well-defined assumptions (Zhou et al., 2022). However, historical precedent suggests that such systems tend to fail more gracefully under distribution shift, whereas specialized pipelines often fail catastrophically when their assumptions are violated (Miller et al., 2021; Jin & Lee, 2025). For agents operating in open-ended settings with evolving tools, tasks, and objectives, the cost of brittleness can dominate the gains from narrow optimization. We believe that even if faced by such situations, each solution to such brittleness would aid more scenarios and attract more research effort, making those costs worth it in the long run. Ample research, we posit, should focus on long-term improvement rather than on current productization issues.

**High volume, high Risk.** This highlights the choice between central and distributive development. Hence, we bring this view as important for the debate, although it has both a supporting and opposing side to our position. Centralized development leverages "many eyes" to standardize code, reduce initial errors, and ensure a coordinated response to vulnerabilities as similar copies can be widely deployed (Raymond, 1999; Goldsmith & Eggers, 2005), where distributed development is built to avoid forced updates (Baran, 1964; Singer & Friedman, 2013). Ultimately, centralization yields fewer errors, but each can be system-wide with large effects.

## 8. Building an Ecosystem for General Agents

We have portrayed above the goal of developing more general agents and discussed the existing and future state of generality. In this section, we enumerate several aspects that we see as necessary steps for the ecosystem as a whole to progress forward (e.g., evaluation and protocols). This section begins with the basic building blocks that are already maturing, but tests minimal generality towards more ambitious endeavors that convey greater generality.

### 8.1. Agentic Skills

A general-purpose agent must possess core competencies such as reasoning, planning, and tool calling, skills that form the common architecture of contemporary agents (Yao et al., 2022; Wang et al., 2023b). These abilities are typically evaluated through static benchmarks where a textual prompt requires a specific response. For instance, GSM8K (Cobbe et al., 2021) assesses step-by-step mathematical reasoning, HotPotQA (Yang et al., 2018) tests multi-hop question answering, and BFCL (Patil et al., 2023) measures tool-use proficiency. Because these benchmarks bypass the need for an interactive loop, responses can be assessed in-

dependently. Consequently, while these foundational skills remain essential for future development, they are already well-represented in current evaluation frameworks.

## 8.2. Interacting with a Dynamic Environment

Agents are also expected to interact with their environment. Those might be stateful and complex: while an LLM can apply tools and receive different responses each time, an ideal agent coordinates tool use so they interact with each other and the environment to accomplish a goal.

Such environments are commonly built for evaluation and can be quite diverse. These include Tau-Bench (Yao et al., 2024; Barres et al., 2025) for customer-service scenarios, AppWorld (Trivedi et al., 2024) for multi-application tasks, WebArena (Zhou et al., 2024) for browser interactions, TerminalBench (Team, 2025) for Linux command-line tasks, and SWE-Bench (Yang et al., 2025) for solving GitHub issues. Each such environment is accompanied by tools and a benchmark for testing. Further generalizations may call for environments where different agents interact (such as in gameplay (Guertler et al., 2025)), and less strict environment-agent coupling, which we discuss next.

## 8.3. Agent Agnostic Evaluation

Evaluation sets the goal for progress, making it crucial in directing development. Thus, when benchmarks assume agents are designed uniquely for them, they implicitly signal that agents must be specialized, not general.

We provide several examples where benchmarks implicitly assume the tested agent possesses certain built-in, domain-specific capabilities, preventing general agent development and hindering comparisons across benchmarks (Bandel et al., 2026; Lacoste et al., 2026) (See also §A). For example:

**Tau-Bench** (Yao et al., 2024; Barres et al., 2025) assumes an agent that can inherently message or converse with a user

```
class BaseAgent(ABC, Generic[AgentState]):
    """Base agent class."""
    @abstractmethod
    def generate_next_message(...,
        message: UserMessage | ToolMessage)
        -> AssistanceMessage:
        ...
```

**WebArena** (Zhou et al., 2024) assumes an agent whose entire perceptual and action space is mediated through a browser interface:

```
class Agent:
    def next_action(..., trajectory: Trajectory)
        -> Action:
        # ActionType: SCROLL, KEY_PRESS,
        # MOUSE_CLICK, GOTO_URL, ...
        ...
```

**TerminalBench** (Team, 2025) assumes an agent whose interaction interface is a computer with a command line:

```
class BaseEnvironment(ABC):
    @abstractmethod
    async def exec(..., command: str)
        -> ExecResult:
        """Executes a command..."""
```

These assumptions are mutually incompatible. A web-browsing agent cannot converse with a user, while a chat-oriented agent cannot click on a web element. Such rigid, benchmark-specific communication protocols make cross-environment evaluation impossible without substantial ad hoc engineering (See illustration in Fig.1).

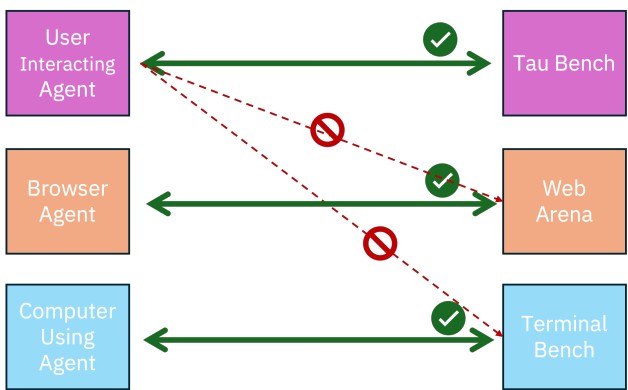

*Figure 1.* Illustration of agents in different benchmarks and their incompatibility with the environment constraints of other benchmarks. Tau-Bench assumes user messaging, which does not align with TerminalBench or WebArena.

A few more general setups can evaluate the same agent on multiple environments without tailoring the agentic pattern, protocol, or benchmark to each. Some works follow this approach: BrowserGym (de Chezelles et al., 2025) standardizes browser interactions across web tasks, and Harbor (Team, 2025) provides a unified terminal protocol. By using a standard protocol, agents can be consistently evaluated across environments, enabling comparisons across models and agent architectures.

Yet, current setups still impose a specific interaction mode. Agents built on different protocols cannot be evaluated natively and must be adapted, creating a mismatch between their intended deployment and evaluation. For instance, MCP-based agents such as Claude Code (Anthropic, 2023) cannot be assessed as they are meant to operate.

## 8.4. Specifying the Environment's Affordances

Just as evaluation should assume little about the agent, a general agent should assume little about the environment. Thus, it must receive or collect the affordances of an unseen environment: what to expect, what can be used, and how it is expected to act. This requires not only innovative proto-

cols but also the ability to learn and retain meta-information about new environments. Most environment adaptation research focuses on open-world exploration (Feng et al., 2024; Anokhin et al., 2024; Ma et al., 2025) or world modeling (Chae et al., 2024), yet rapid adaptation and learning to adapt across environments remain underexplored. Part of it, might be a lack of fitting benchmarks. Current benchmarks rarely define, in an agent-agnostic way, the tasks, observations, or action effects, forcing developers to infer or invent these elements, which limits systematic progress.

Some benchmarks force the agent to be specific to the environment. For example, SWE-Bench (Yang et al., 2025) defines the problems agents should solve but provides no standard instructions for solving, validating, or structuring solutions. Thus, rewarding agents tailored to the benchmark, while penalizing general deduction.

Other Benchmarks make the environment agent-specific. For example, Tau-Bench (Yao et al., 2024; Barres et al., 2025) that provides environment-level specification but assumes a specific LLM agent is used (See App. B and §8.3).

Across both cases, the core issue is the same: We lack a standardized, agent-agnostic interface that cleanly communicates the task, available information, and action space.

## 8.5. Standardization

A common driver for developing new and general technologies is standardization. We find ample room for standardization in the field and provide several examples in this section. Arguably, this represents the sweet spot of work, both lacking and with mature enough infrastructure to immediately build upon and see adoption.

Standard communication protocols offer great potential for development. For example, by providing a common interface for agent-environment communication or separating Currently, as mentioned above, each environment or benchmark requires custom integration scripts, unique logging conventions, and bespoke output formats. However, despite advances in agent frameworks and protocols, such as MCP (Anthropic, 2024) and A2A (Developers, 2025), there is still more to be done (See App. C).

Evaluation and analysis represent further areas where standardization is critical. Current benchmarks lack consistency in how metrics are defined, labeled, and aggregated; furthermore, the reporting of computational costs is often omitted or poorly documented(Perlitz et al., 2024; Ndzomga, 2026; Ghosh et al., 2026) (More discussion in App. C). Moreover, establishing uniform logging formats would facilitate systematic failure analysis and the identification of performance bottlenecks, ultimately enabling more precise refinements to reasoning and decision-making modules.

Finally, standard protocols provide safety and reliability benefits as a natural byproduct. They allow interventions and reduce unpredictable behavior by defining clear rules for agent interactions, error handling, and escalation. Detailed logs further support retrospective abilities for debugging, auditing, and compliance, ensuring that agents operate within defined boundaries. Critically, these safety guarantees do not come at the cost of development speed, allowing teams to innovate while maintaining predictable and trustworthy behavior.

## 8.6. Compositionality

A more ambitious yet high-potential objective is compositionality. Achieving a modular framework would allow researchers to deploy identical agentic patterns across models and toolsets with minimal reconfiguration. Such decoupling facilitates clearer credit assignment, as it isolates the performance contributions of individual components, thereby enabling independent optimization of each layer (see the "narrow waist" discussion in §6).

Some frameworks already strive towards compositionality, such as HAL (Kapoor et al., 2025a), which compiled nine domain-specific benchmarks. Each environment in HAL comes allows switching models, using the same agentic pattern. It does not, however, allow for adding general agents and communicating the necessary information with them (See §8.4).

## 8.7. Modalities, and Embodiment

While language remains the cornerstone of agentic systems— often augmented by vision (Niu et al., 2024; Schumann et al., 2024)—the next frontier lies in integrating diverse perceptual modalities, novel sensing, and direct physical control. Early developments in robotics and embodied AI (Wang et al., 2023a), alongside emerging foundation models in the sciences (Van Breugel & Van Der Schaar, 2024; Dallatorre et al., 2024; Batatia et al., 2023), signal a departure toward agents that transcend traditional computer interfaces. We advocate for expanding the operational scope of agents to include heterogeneous foundation models and complex physical environments. Rather than prescribing specific solutions, such as whether protein models should be treated as modular tools or integrated protocols, we call for research that identifies generalizable principles across these diverse domains, seeking a unified research agenda.

# 9. Conclusion

Current research and evaluation practices largely favor agents that are tightly coupled to specific benchmarks, tools, or domains. We argued that such specialization is a transitional stage, and that progress should increasingly prioritize

general-purpose designs that can operate across unfamiliar environments with minimal manual intervention.

Realizing this potential demands an ecosystem-level effort. Benchmarks, protocols, and evaluation frameworks must all pivot from encoding agent-specific assumptions to explicitly describing environment affordances and interaction semantics. While specialized agents will continue to play an important role, we contend that investing in general-purpose agentic systems provides a stronger foundation for scalable, adaptable, and responsible AI agents operating in open-ended real-world settings.

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

| Level | Cross-model | Interaction | Cross-env. | Cross-agent | Prot.-agnostic | Examples |
|---|---|---|---|---|---|---|
| 1: Agentic skills | Yes | No | No | No | No | BFCL, GSM8K |
| 2: Interactive model | Yes | Yes | No | No | No | Tau-Bench, WebArena |
| 3: Cross-model harness | Yes | Yes | Yes | No | No | HAL |
| 4: Protocol-centric | Yes | Yes | Yes | Yes | No | BrowserGym, Harbor |
| 5: General evaluation | Yes | Yes | Yes | Yes | Yes | (missing) |

*Table A.1.* Comparison of five levels of agent evaluation, highlighting cross-model, cross-environment, cross-agent, and protocol coverage.

## A. The generality of different evaluation benchmarks.

In Table A.1 we provide a classification of different benchmarks and how general they are.

Cross-model means that the can evaluate multiple agents (or models in the agentic skills case), whether they test interactiveness, multiple environments, multiple general agents, and whether those agents can operate in the way they expect or need to be adapted to the benchmarks.

## B. An example from Tau Bench.

We bring an example of how environment details are shared with the agents, but it is done through an instruction to the agent, assuming this is how the agent operates, and not, for example, through API calls.

```
AGENT_INSTRUCTION = """
You are a customer service agent that helps the user according to the <policy> provided below.
In each turn you can either:
- Send a message to the user.
- Make a tool call.
You cannot do both at the same time.

Try to be helpful and always follow the policy. Always make sure you generate valid JSON only.
"""
```

While this instruction block expresses important aspects of the task, it mixes them with assumptions specific to a particular agent design—such as turn-by-turn interaction rules, usage of tools and JSON output formatting. These constraints make sense for a conversational LLM but do not generalize to other types of agents, such as code-act agents.

## C. More on Protocol Standardization

We expand on examples where current protocols may be more standard and refer to (Bandel et al., 2026; Lacoste et al., 2026) for more discussion. We consider the common MCP protocol as a case study for whether existing protocols suffice for developing general agents.

MCP defines three core primitives: tools (invocable operations), resources (exposed data/content), and prompts (parameterized templates), along with mechanisms for streaming events and updates. While MCP provides a promising foundation for unifying agent-environment interaction, several gaps prevent it from serving as a complete solution for general-agent evaluation:

- **Missing Support for Benchmark Task Semantics.** Benchmarks center around tasks—the defined goal an agent is supposed to achieve. MCP does not offer a built-in way to represent or communicate such tasks. One could require that tasks always appear in either prompts, resources, or events, but doing so would essentially create a new protocol on top of MCP. Lacking a way to communicate it, we assume the task is somehow manually defined.

- **Missing Support for Evaluation Workflows.** Evaluation requires more than interaction; it depends on standardized metrics reporting, aggregation, logging, experiment tracking, and reproducibility. MCP is intentionally agnostic to these needs.

- **Inconsistent Ecosystem Adoption.** Support for MCP remains uneven: many frameworks implement tool calling but not resources or prompts, resulting in inconsistent capabilities and substantial integration overhead (Table C.1).

These limitations indicate that while protocols like MCP and A2A lay important groundwork, they do not yet meet the full requirements of standardized general-agent evaluation.

| Agent / Framework | Tools | Resources | Prompts |
|---|---|---|---|
| Smolagents | Yes | No | No |
| Llama Stack | Yes | No | No |
| OpenAI Agents SDK | Yes | No | Yes |
| Codex CLI | Yes | No | No |
| Claude Code | Yes | Yes | No |

*Table C.1.* MCP integration across common agent frameworks. In many cases, only partial protocol components are implemented.

| Metric type | Tau-Bench | AppWorld | SWE-Bench |
|---|---|---|---|
| Success (bool) | Reward = 1 | success | Resolved |
| Score (float) | Reward | Score | – |
| Termination | Termination reason | – | – |
| Duration | Duration | – | – |
| Interactions count | Num messages | Steps | – |
| Agent cost | Agent cost | – | – |
| Environment cost | User cost | – | – |
| Task ID | Task ID | Task ID | Instance ID |
| Logs | Message log | Task logs | Test logs |
| Success rate (agg) | Avg reward | Goal completion | Resolved counts |
| Cost aggregate (agg) | Avg agent cost | – | – |

*Table C.2.* Metrics differ across benchmarks, with incompatible names and formats even for basic success and cost reporting.

## C.1. Standardization to Ease Development

Just like the agents that require standard specification, so do researchers and developers. To support seamless experimentation—without hours spent integrating each new environment—researcher-facing interfaces must also be standardized and simplified. Today, every environment demands its own setup, scripts, and output formats. Researchers repeatedly lose time to these inconsistencies and risk introducing avoidable integration errors. This was done well in various non-agentic contexts, such as specifying hundreds of games (Guertler et al., 2025) or Reinforcement Learning Environments (Brockman et al., 2016) in the same format.

One example of this fragmentation is the process of collecting and interpreting results. Benchmarks report outcomes in different formats, store them in different locations, and follow different conventions. As shown in Table C.2, even the basic metrics differ across three representative benchmarks—not only in which quantities are tracked, but also in terminology and aggregation standards used.

A simple notion like success appears as a Boolean *resolved* flag in SWE-bench, a *success* field in AppWorld, and an implicit *reward of 1* in Tau-bench. Interaction cost is reported as *agent cost* and *user cost* in Tau-bench but omitted entirely in AppWorld and SWE-bench. Even when benchmarks measure conceptually similar quantities, they adopt incompatible formats, naming conventions, and aggregation methods.

These gaps underscore the need for consolidation and standardization to make large-scale evaluation of general agents feasible. This is not just a convenience—given the growing complexity of modern benchmarks, it is the only viable path to scalable general agent research.

