# OpenReview forum: "Position: Agentic Systems Should be General"
_ICML.cc/2026/Position_Paper_Track — ICML 2026 Position Paper Track regular_

### Official Review · Reviewer_Myx8 · 2026-02-24

**Significance:** 2
**Argument Clarity:** 2
**Ethics Flag:** Yes
**Rating:** 2
**Confidence:** 4

**Questions:**

(1) Regarding novelty, who in the community is arguing against generality, and what specific research decisions would change if your position were adopted?

(2) Regarding the roadmap, can you provide a priority ordering among the ecosystem-building steps in Section 8, clarifying which are most urgent and which are prerequisites for others?

(3) Regarding empirical evidence, how do you address the confound that the general and specialized agents in your comparisons use different foundation models, and do you have evidence from more complex settings beyond text/code environments?

(4) Regarding alternative perspectives, have you considered whether pursuing general agent components rather than general agent systems might be a more effective path, and how do you address the deployer preference for reliability over generality?

(5) Regarding ethics, how do you reconcile advocating for a centralized ecosystem with the risks of monopolization and single points of failure, and how do you address the dual-use concern that lowered entry barriers also benefit malicious actors?

**Alternative Views Section:**

Yes

**Compliance With Llm Reviewing Policy A Conservative:**

Affirmed.

**Discussion Potential:**

2

**Ethical Review Concerns:**

The paper advocates for a centralized general agent ecosystem and presents lowered entry barriers as a benefit, but does not discuss the risks of monopolization and single points of failure from centralization, nor does it address that lowered barriers equally enable malicious actors to misuse general-purpose agents across diverse environments at scale.

**Ethics Review Area:**

["Inappropriate Potential Applications & Impact (e.g., human rights concerns)"]

**Paper Summary:**

This paper argues that agentic systems should prioritize general-purpose designs over benchmark-specific solutions, supported by ecosystem-level improvements in protocols, evaluation, and standardization. The authors analyze benchmark-conditioned agents' limitations, present two cases where general agents approach specialized performance at lower cost, and draw on historical trends toward abstraction.

However, the position restates a widely shared aspiration, the roadmap lacks prioritization, the empirical evidence is limited and confounded, and discussions of alternative perspectives and ethical risks remain insufficient.

**Position:**

Yes

**Position In Title:**

Yes

**Related Work:**

2

**Strengths And Weaknesses:**

weakness:

- **1. Limited Novelty: Pursuing Generality Is Not a New Position**

Pursuing general-purpose agents has been a mainstream direction in the AI community. From GATO to Voyager, from OpenAI's general tool use to Google's multimodal agents, both industry and academia have been actively working toward generality. The paper itself acknowledges that early work like GATO already pursued broad generality. The paper attempts to distinguish its contribution by criticizing "benchmark-conditioned" pseudo-generality (e.g., HAL's if-else branching), but this reads more as an engineering observation than a stance requiring a position paper. The community is not opposed to generality; rather, it is making pragmatic tradeoffs between generality and practical performance. The authors need to more clearly articulate who is arguing against generality and what specific research decisions their call would change.

- **2. Overly Broad Roadmap with Limited Actionability**

The ecosystem-building agenda in Section 8 covers agentic skills, dynamic environment interaction, evaluation, environment affordances, standardization, compositionality, multimodality, and embodiment, spanning nearly every direction in agent research. This reads more as a survey of the entire field's research agenda than a focused action plan. The paper does not provide a priority ordering: which steps are most urgent? Which are prerequisites for others? For instance, should protocol standardization precede evaluation frameworks? What are the dependencies between environment affordance specification and compositionality? Without such structured prioritization, readers gain limited actionable guidance from the roadmap.

- **3. Weak and Selective Empirical Evidence**

The paper uses two cases (ASTA Bench and SWE-Bench) to support the viability of general agents, but the authors themselves acknowledge this evidence is "anecdotal." More importantly, both cases involve relatively structured, text/code-based environments where general agents perform close to specialized systems. The paper does not demonstrate whether general agents can maintain such competitiveness in more complex settings (e.g., embodied, multimodal, or real-time interactive environments). Additionally, comparing general and specialized agents using different foundation models (GPT-5 vs. Claude 4 Sonnet) is not entirely fair, as differences in model capability may confound the contribution of generality itself.

- **4. Missing Alternative Perspectives**

While Section 7 provides a reasonable discussion of alternative views, some important perspectives are absent:

(a) Generality may be the wrong level of abstraction. Perhaps the goal should not be general agent systems, but general agent components (e.g., general planners, general tool-calling interfaces) that are composed into domain-specific systems. The paper mentions compositionality but does not seriously discuss whether this is a better path than pursuing end-to-end general agents.

(b) The user and deployer perspective is missing. The paper argues for generality entirely from the researcher and developer viewpoint. In practice, users and enterprises may value reliability, auditability, and customizability over generality. A well-validated specialized agent in a specific domain may be more trusted than a theoretically more general but less thoroughly tested system.

- **5. Insufficient Discussion of Ethical Issues**

The paper briefly mentions safety, governance, and accountability in Sections 6 and 7, but these discussions are overly optimistic and lack depth:

(a) Centralization risks. The paper advocates for a centralized general agent ecosystem but does not discuss the resulting risks of monopolization, single points of failure, or suppression of diversity. If the entire community develops around a few general agent frameworks, a flaw in one framework could affect all downstream applications.

(b) Differential impact on stakeholders. While the paper presents lowered entry barriers as a benefit, it does not discuss that this equally lowers the barrier for malicious actors, making misuse easier alongside legitimate use.

**Support:**

1

---

> ### Author Rebuttal · Authors · 2026-03-30
>
> We acknowledge that our position aggravated the reviewer, while we did not intend to; it might at least show that this paper is divisive and worth a discussion.
> Regarding arguments against generality. First of all, the reviewer themselves, debate against it, stating the centralization and other aspects might have negative implications, this is the debate we want to raise. We had similar debates among the group of writers and outside it, so we are surprised this is raised as an issue. Concretely, as our larger field is not a philosophy field, most of the arguments are found in actions toward a certain direction or another, not in arguments against a point. Thus, while the reviewer mentions some of our examples of papers that do go in that direction, there are many more going against it. For example, consider the amount of code specific benchmarks and papers. To add to it, during the writing of the paper, many debated against it and even saw it as unrealistic.
>
> Indeed, we chose to cover broadly where the field can take actions upon our suggestions, we propose a position, not a specific plan actionable by one group. As such, it will be developed in many directions by a whole community, and we dedicated 2.5 pages to those. However, this complaint is also unfair as we do focus on the more immediate steps, such as generalizing over environments and protocols for communicating the task the agent needs to perform. Thus, 8.6 about embodiment is a single paragraph and focus areas like evaluation in 8.3 cover almost a full page. Will adding a concrete prioritization or our assessment of impact and effort required for each direction satisfy this? If not, how would you imagine it proposed? We are happy to add more suggestions to the camera ready and expand on the proposed priority if we had to choose (and we won’t, most will be on parallel).
>
> As the reviewer and us both agree, there are only hints of general agents now (going back to the novelty point) the evidence that those are worthwhile is promising, but as we both state, anecdotal. Thus, we provide such evidence as well as an argumentation fitting the call for papers: “Support: The paper supports its position with clear reasoning and evidence where appropriate.“. Note this is also a minority view, other reviewers accepted, we do it. If the reviewer could provide concrete feedback we can act upon rather, it would be useful for improving our paper.
>
> Regarding compositionality vs. end2end, there are different ways where the same goals may be achieved, in this work we do not want to prescribe which technology will eventually work best and this is up to the community to invent and prove empirically. We can state it in the paper.
>
> Indeed, we agree that this paper addresses the research and development community rather than practitioners and deployers, this is not a weakness but a fact. Practitioners, at any given time, will want to use the best current technology for specific causes, which, as we state and will emphasize, is not general and will include employing dedicated systems. The (ideally general) technology developed by the community of course will want to have other traits except just working, such as being reliable, auditable, safe, etc.
>
> Regarding the safety aspects, we discuss it in 6 paragraphs, but the point of a failure that goes across system is a known that comes with the ability to solve it in one place which we discuss. We will raise it as a potential issue.
>
> We do not discuss that allowing people to use a technology means this is true for good and bad people alike; this is a common truth and not specific to this case, so if one pursues this technology at all (a wonderful question to ask IMHO, but not relevant to our paper), they should expect it to be used. Do you think it is not obvious? We can add it.
>
> Also, regarding question 3. We did not understand it. Yes, those are different systems, but the cost is not only due to the model but also how it is being used; separating model and agentic pattern effects is empirical work worth doing, which some of the papers we cite indeed pursue. But what complex settings are more than text and code? You mean visual agents as well? Code and browsers and such are by far the most common domains addressed; this is the whole point of this position…

---

### Official Review · Reviewer_3pnL · 2026-03-11

**Significance:** 3
**Argument Clarity:** 3
**Rating:** 4
**Confidence:** 2

**Questions:**

Do you have plans to run more systematic evaluations across diverse benchmarks?

**Alternative Views Section:**

Yes

**Compliance With Llm Reviewing Policy A Conservative:**

Affirmed.

**Discussion Potential:**

3

**Final Justification:**

The rebuttal from the author addressed my concerns to some extent, so I raise my rating.

**Paper Summary:**

This paper advocates for a shift in AI agent research: moving away from systems over-optimized for specific environments and toward truly general-purpose agents. The authors figured out a fact that current systems are often tightly coupled to benchmarks, which fiercely limits their ability to generalize out-of-domain. To support this, the authors compare benchmark-specific setups with environment-agnostic approaches. Using a few case studies, they demonstrate that simpler, generic action loops can often perform just as well without the bloated, environment-specific logic. The paper concludes with a call to action for modular architectures, standard interfaces, and better evaluation protocols that actually reward generality rather than benchmark hacking.

**Position:**

Yes

**Position In Title:**

Yes

**Related Work:**

3

**Strengths And Weaknesses:**

Strengths：
1.Tackles a very timely and frustrating issue in current agent research: benchmark overfitting vs. actual generality.
2.Draws a crisp, useful line between benchmark-conditioned and environment-agnostic systems.
3.Good high-level discussion on misaligned research incentives and flawed evaluation protocols in the broader ecosystem.
4.The concrete examples used to ground the argument are well-chosen and helpful.

Weaknesses:
1.The empirical backing is fairly thin; the evidence feels more anecdotal than systematic.
2.Heavy on critique, light on solutions. The paper lacks actionable, concrete design proposals.
3.Glosses over the actual trade-offs—specialization isn't always bad, but this nuance is largely missing.
4.The literature review feels sparse and could connect more deeply with existing work.

**Support:**

3

---

> ### Author Rebuttal · Authors · 2026-03-30
>
> Thank you for the overall positive review stating that we suggest a clear cut point that would be interesting to discuss.
>
>
> We note a strong difference between a position paper and a regular one. Some of us are working on more systematic evaluations across benchmarks (as you can see in our citations and supplementary materials, acknowledging that). Still, this is a position paper, one that should not commit to a single solution. A common, but boring position framing goes along the lines of “this is the right thing to build, let someone else build it”. Instead, we pose a broad view for the community, “here is a direction we should build more on, but one paper cannot do it all, let’s build it together”. We hope this paper will be relevant for discussion and to ponder on when researchers consider their future work. Thus, we make it a point not to dictate a single set of steps that we can just pursue, but make a position that is somewhat divisive or highlighting the difference between approaches one may choose to take. We do, however, in the last two and a half pages, give not solutions but directions on where, in practice, this position can matter and what is worth pursuing more than other things. Given the above perspective, what kind of actionable proposal would you find useful? We are happy to think and discuss deeply and add a bit more of that, if you point us to what you think would really make a difference, pushing more researchers to think and debate it (and even disagree), or change the original plan they had.
>
> Regarding the literature review, what are you referring to? Overall, we had approximately 75 papers cited by us. What areas, papers, or claims you see as missing? We tried to be gracious and welcome any suggestion to expand.
>
> A sidenote, while you fairly say we “critique”, while we might disagree with general trends, and take a strong stance, maybe stronger than some of us would even hold, to give an interesting and defensible position. We have deep respect for the research in the field, its fast pace and major breakthroughs. If you feel that somewhere, where we aimed at raising discussions we were unfair, please let us know as well.
>
> Regarding the claim that specialization is always bad. It most definitely is not. In fact, we think it is practically very useful. There are two points about it. One, that practitioners would always care about their special usecase, for a good reason, we want to give them methods that provide the least efforts for best results (i.e. general). We stated such points, for example in the subsection “Better starting point for Specialized Agents”, but rereading it, after various edits this point became a implicit. We will emphasize (see bellow)
>
> Moreover, we set a binary position stating that researchers should make (more) general solutions. We do not think the field would just abandon specialized work completely, this is unrealistic and probably unwanted. But, calling for a balanced amount of work on general and non-general where currently non-general is underresearched is uninteresting intellectually and won’t leave place for the discussion on the advantages we do see for pursuing generality, those unconvinced will be the ones to not pursue it.
>
> We accept this criticism, and are happy to put forth those nuances into the paper. We have two follow-up questions here (on the two paragraphs above). (1) Do you think something of the reasoning for putting forth a binary distinction worth mentioning? For example, conclude with this and a few other such decisions we needed to make to focus the argument in the paper? (2) Surely, we will emphasize more that we want (also) specialized systems and the question is the right long term path for method development. Do you think that would be satisfactory, would you add another nuance in this issue for the final version?

---

> > ### Author Rebuttal · Reviewer_3pnL · 2026-04-03
> >
> > The authors have addressed my concerns and I will raise my score.

---

### Official Review · Reviewer_wGaq · 2026-03-12

**Significance:** 4
**Argument Clarity:** 4
**Rating:** 5
**Confidence:** 3

**Questions:**

Q1. This paper argues for agent-agnostic evaluation across different environments. What are the minimal requirements for such an evaluation framework or protocol in the near term?

Q2. If we can define transformations between different protocols and show functional equivalence, can those protocols be considered agent-agnostic in practice?

**Alternative Views Section:**

Yes

**Compliance With Llm Reviewing Policy A Conservative:**

Affirmed.

**Discussion Potential:**

4

**Final Justification:**

I am aware of other reviewers' opinion. I maintain my current rating.

**Paper Summary:**

This paper argues that agentic systems should be designed as general-purpose systems that can adapt across diverse and unfamiliar environments, rather than being optimized for narrow benchmarks. It frames current domain-specific agents as a transitional stage and advocates a long-term shift toward unified, adaptable agents that handle varied tasks, tools, modalities, and settings with less per-instance human intervention. The paper’s main contributions are conceptual: it defines core agentic-system terms, formalizes generality as a spectrum (including the idea that higher generality means lower manual deployment effort), and emphasizes that generality must still be paired with strong practical performance. Its central position is that the community should prioritize protocols, evaluation frameworks, and development practices that reward adaptiveness and support a shared ecosystem for general agentic systems.

**Position:**

Yes

**Position In Title:**

Yes

**Related Work:**

3

**Strengths And Weaknesses:**

**Strengths**

S1. The paper clearly points out that most existing agentic systems are still designed for specific domains or sets of tasks, and that current benchmarks are not sufficient for evaluating truly general agentic systems. This provides a reasonable motivation for the paper's position and is well aligned with the current state and likely future direction of the field.

S2. The discussion of alternative views is thorough and thoughtful, and it helps strengthen the paper's central position.

S3. Section 8 presents a useful outline for building an ecosystem around general agents. This discussion helps illustrate the broader research agenda and may encourage further work toward this goal.

S4. The paper is well organized and easy to follow.

**Weaknesses**

W1. One potential limitation is that it remains unclear how to build a valid and practical benchmark for assessing general agentic systems. Although evaluation aspects can be diversified, they are still finite. If the goal is strong generalization to new environments, performance should also be validated on genuinely unseen tasks.

**Support:**

4

---

> ### Author Rebuttal · Authors · 2026-03-30
>
> Thank you for the deep review and for capturing the nuance of our work, from the working definition to generality to the conceptual contributions.
>
> As this is a position paper, and a benchmark would require a whole paper to describe, explain specific decisions, evaluations etc., what do you think would have been helpful? Two parts of our group are indeed thinking and building more general evaluation approaches (which we acknowledged according to ICML cross-submission guidelines), so we have a lot to say, but we do not want to prescribe our approaches, those are just the first. A position paper should cause a debate or make someone invent their own better approach. With some clarity of what to add there, we will gladly expand on this point. If those are Q1,Q2, we can add discussion of it to the paper.
>
> Q1: In the near term, probably something that can run all or most agentic benchmarks on all or every agent without changing anything in the agent or benchmark between runs (but probably setting some protocol layers in between). This is along the lines of your Q2 as well, right?
> Q2: Yes, that sounds totally fair and quite a good approach, few people already have initial works in those directions. If you do that than you can always set some code to move between them. Assuming you know in advance when to apply this transformation (e.g. every time I use Agent A).

---

> > ### Author Rebuttal · Reviewer_wGaq · 2026-04-03
> >
> > Thanks for your response. I will maintain my rating to Accept.
> >
> > (1) For Q1, you may include some of the efforts from your group to show that the proposal is practical and ongoing.
> >
> > (2) For Q2, you may argue in your revised paper that existing benchmarks, including Tau-Bench, WebArena, and TerminalBench, are not equivalent to agent-agnostic evaluation protocols. Hence, the concrete proposal of your paper is necessary.

---

### Official Review · Reviewer_jnqY · 2026-03-13

**Significance:** 3
**Argument Clarity:** 3
**Rating:** 4
**Confidence:** 2

**Questions:**

1. The paper emphasizes the importance of generality, but how should generality be formally measured? Are there specific metrics or benchmarks the authors envision for evaluating agent generalization across environments?
2. How should researchers balance the potential benefits of generality with the efficiency advantages of specialized systems optimized for specific domains?

**Alternative Views Section:**

Yes

**Compliance With Llm Reviewing Policy A Conservative:**

Affirmed.

**Discussion Potential:**

3

**Paper Summary:**

This paper argues that the machine learning community should prioritize general-purpose agentic systems that can operate across diverse tasks and environments rather than systems specialized for individual benchmarks or domains. Agentic systems are defined as frameworks combining foundation models, tools, and execution strategies to interact with external environments.

A broad area investigated by the paper is the development of agents capable of adapting to unfamiliar environments with minimal human intervention. The authors argue that current agent research often embeds benchmark-specific assumptions into agent logic, which limits generalization. They advocate for environment-agnostic agent designs, standardized protocols, and agent-agnostic evaluation benchmarks.

**Position:**

Yes

**Position In Title:**

Yes

**Related Work:**

4

**Strengths And Weaknesses:**

Strengths
1. The paper clearly states and consistently advocates its central position: that the research community should prioritize the development of general-purpose agentic systems. The argument is presented explicitly and reinforced throughout the paper.
2. Agentic systems have recently become a central focus of machine learning research. The question of whether the field should emphasize general-purpose agents or specialized task-specific agents is both timely and important.
3. The paper provides examples illustrating how simple, environment-agnostic agents can achieve competitive performance relative to complex specialized systems. These examples help motivate the argument for generality.

Weaknesses
1. Although the paper discusses generality extensively, the concept remains somewhat loosely defined. It is not always clear how generality should be measured or evaluated in practice.
2. While the paper discusses potential drawbacks of specialized agents, it could further explore the trade-offs between generality, efficiency, safety, and predictability.

**Support:**

3

---

> ### Author Rebuttal · Authors · 2026-03-30
>
> We thank you for the careful read and support of why this debate is relevant and interesting.
> As to your questions:
> 1. Regarding generality, (as Reviewer wGaq puts well) we formalize it “as a spectrum (including the idea that higher generality means lower manual deployment effort)”. Thus, as we state in 4.1, this kind of generality goes beyond OOD that AI traditionally faced, and measuring it is not as straightforward; moreover, it is a measure for how general the approach is, not how well is the system generalizing. For a given level of generalization required (e.g. OOD, environment etc.), one can create benchmarks that differ and see whether the model succeeds. In other words, for each progress in more general agents, we expect one of the first things to be, as you suggested, setting a target (benchmark and metric), for example, benchmarks that provide the model with the environment and task details as inputs as you stated we expand on (partially because it is a concrete next step). We will pass on the places where we discuss it and make sure this distinction is emphasized.
> 1. We were discussing this point of specialization and balances a lot among ourselves. For practitioners, each group would likely benefit most from taking the best existing technology and specialize it for their own needs. In contrast, as a research field, such specializations have the most benefit when they offer general understanding of how to specialize to other cases as well, and this can be impactful research. We argue for mainly investing in general improvements, that would serve as better starting points for general use, as well as for specialization over them. In that sense, the tradeoff is nuanced. It is in some sense less of a tradeoff and more of a reprioritization of what technologies to focus on as a field, ones others will be able to use, for their own needs with little effort, or specialize well with minimal effort. Does that make sense? We discuss it in the paper, but maybe it is worth expanding on a bit in the camera ready. A side note is that of course we believe some research should be dedicated to both, but this is beyond the position stating we should pursue general ones.
> 1. Regarding the question part on safety and other characteristics of the tecnology, of course this research would continue to be an important part of the field going forward, regardless of which other goals are pursued, and it will need to support general methods that ensure safety, rather than for example code safety only methods. For example through human interventions self regulation etc.

---

### Decision · Program_Chairs · 2026-04-30

**Decision:**

Accept (regular)

**Comment:**

Reviewers were mixed in their appraisal. Most reviewers found the core position to be extremely timely and clearly articulated. Furthermore, reviewers appreciated the discussion of alternative views

At the same time, key weaknesses were pointed out by several reviewers, including (1) Limited Novelty: pursuing generality is not a new position, (2) Limited Actionability: it remains unclear how to translate the proposal into action (with several reviewers critiquing the lack of a consensus definition for what generality means here), and (3) Glosses over the trade-off: there is a nuanced tension between specialization and generality that is under-discussed here.

Overall, it's a borderline appraisal, with the weaknesses outweighing the strengths. I encourage the authors to take the reviewers comments into account as they further their work.